# Allelopathic Potential of the Cyanotoxins Microcystin-LR and Cylindrospermopsin on Green Algae

**DOI:** 10.3390/plants12061403

**Published:** 2023-03-22

**Authors:** Ivanka Teneva, Violeta Velikova, Detelina Belkinova, Dzhemal Moten, Balik Dzhambazov

**Affiliations:** 1Faculty of Biology, Paisii Hilendarski University of Plovdiv, 4000 Plovdiv, Bulgaria; 2Institute of Plant Physiology and Genetics, Bulgarian Academy of Sciences, 1113 Sofia, Bulgaria; 3Institute of Biophysics and Biomedical Engineering, Bulgarian Academy of Sciences, 1113 Sofia, Bulgaria; 4Institute of Biodiversity and Ecosystem Research, Bulgarian Academy of Sciences, 1113 Sofia, Bulgaria

**Keywords:** cyanotoxins, microcystin-LR, cylindrospermopsin, allelopathy, green algae, *Chlamydomonas*, *Dunaliella*, *Scenedesmus*

## Abstract

Allelopathic interactions are widespread in all aquatic habitats, among all groups of aquatic primary biomass producers, including cyanobacteria. Cyanobacteria are producers of potent toxins called cyanotoxins, whose biological and ecological roles, including their allelopathic influence, are still incompletely understood. The allelopathic potential of the cyanotoxins microcystin-LR (MC-LR) and cylindrospermopsin (CYL) on green algae (*Chlamydomonas asymmetrica*, *Dunaliella salina*, and *Scenedesmus obtusiusculus*) was established. Time-dependent inhibitory effects on the growth and motility of the green algae exposed to cyanotoxins were detected. Changes in their morphology (cell shape, granulation of the cytoplasm, and loss of flagella) were also observed. The cyanotoxins MC-LR and CYL were found to affect photosynthesis to varying degrees in the green algae *Chlamydomonas asymmetrica*, *Dunaliella salina*, and *Scenedesmus obtusiusculus*, affecting chlorophyll fluorescence parameters such as the maximum photochemical activity (F_v_/F_m_) of photosystem II (PSII), the non-photochemical quenching of chlorophyll fluorescence (NPQ), and the quantum yield of the unregulated energy dissipation Y(NO) in PSII. In the context of ongoing climate change and the associated expectations of the increased frequency of cyanobacterial blooms and released cyanotoxins, our results demonstrated the possible allelopathic role of cyanotoxins on competing autotrophs in the phytoplankton communities.

## 1. Introduction

Cyanobacteria are prokaryotic organisms that can be found in most water bodies [1]. During their intensive growth (so-called “blooming”), many of them are able to cause adverse effects with serious consequences, not only for human and animal health, but also for other phytoplankton organisms living in the same aquatic biomes [1,2]. This is mainly due to a group of secondary metabolites known as cyanotoxins. The cytotoxic effects of these compounds have been well studied in animals and humans [3,4], but there are many questions regarding their effects on the coexisting phytoplankton communities [5,6,7,8,9]. There is still no consensus on the ecological role of the cyanobacterial toxins [6,10,11,12]. This is partly due to the lack of information on the effect of cyanobacterial “blooms” and released cyanotoxins on individual phytoplankton communities in the aquatic ecosystems, and in particular, the allelopathic potential of the known and most common cyanotoxins. Cyanotoxins are thought to have allelopathic potential due to their inhibitory or stimulatory effects on other aquatic organisms [9,13,14,15].

Allelopathy is a biological phenomenon in which biochemicals produced by one organism have positive or negative effects on other organisms. This process is a unique adaptation to gain a competitive advantage over other organisms living in the same community [16]. It is considered as one of the factors that promotes and supports the mass blooming of cyanobacteria and algae in freshwater, brackish, and marine ecosystems around the world [16,17]. Allelopathic interactions play an important role for the structure and biodiversity within a community. Such interspecies or intraspecies interactions can occur in all aquatic ecosystems and among all groups of primary aquatic producers [18]. Given the fact that due to climate change there is an increase in the intensity of cyanobacterial blooms worldwide [9,19,20] it is important to determine the possible allelopathic effects of cyanotoxins and the affected organisms.

Allelochemicals are a variety of bioactive secondary metabolites that are not essential for the metabolism of the allelopathic organism. Only a small proportion of cyanobacterial substances with biological activity can be classified as allelochemicals. These substances include cyclic and non-cyclic peptides, polyketides, alkaloids, phenols, and chlorinated aromatic compounds [15,21,22,23,24]. The mode of action of the allelopathic substances includes growth inhibition, photosystem II (PSII) inhibition, cell motility inhibition, enzyme activity inhibition, cell membrane lysis, reduction in the pigment content, or increase in the oxidative stress [24,25,26,27].

According to most publications on the subject so far, the effects of cyanotoxins on other phytoplankton organisms are associated with the inhibition of phytoplankton growth, the inhibition of chlorophyll production, and the inhibition of photosynthesis (reviewed in [21]). Jüttner and Lüthi demonstrated a lack of such inhibitory effects and so the role of microcystins as allochemicals is disputed [28].

Microcystins (MCs) belong to a family of cyclic heptapeptides with a common structure of cyclo-(*D*-Ala^1^-X^2^-*D*-MeAsp^3^-Z^4^-Adda^5^-*D*-Glu^6^-Mdha^7^), in which X and Z are variable *L*-amino acids, *D*-MeAsp is *D*-*erythro*-β-methylaspartic acid, and Mdha is *N*-methyldehydroalanine [29]. These cyanotoxins are produced by certain strains of planktonic blooming cyanobacteria (*Microcystis*, *Anabaena*, and *Planktothrix*) or terrestrial/benthic cyanobacterial genera (*Nostoc*, *Hapalosiphon*) [1,30]. Microcystins are the most common and intensively studied biologically active substances produced by blooming cyanobacteria. Although they accumulate in the cells that produce them, they can also be found in the environment, even in the early stages of cyanobacterial blooms. Apparently these toxins are released during the lysis of cyanobacterial cells and this happens constantly [31]. However, these toxins are released into the environment not only through aging and lysis of the blooms, but also through an active release, as some researchers have shown [1,32].

Microcystins have been shown to cause adverse allelopathic effects on both higher plants [33,34,35,36,37] and phytoplankton [9,25,38,39]. For example, the three most commonly detected forms of microcystin in water bodies (MC-LR, MC-RR, and MC-YR) cause morphological and physiological changes in *Scenedesmus quadricauda*, inducing cell aggregation, increasing cell volume, and increasing the production of photosynthetic pigments. [31]. Cyanobacteria producing microcystins are thought to have a significant environmental advantage by using these heptapeptides to eliminate their phytoplankton competitors from the environment [9,13,39], but the complex network of interactions at the molecular level remains unclear [9,40].

Cylindrospermopsin (CYL) is an alkaloid comprising a tricyclic guanidino moiety linked via a hydroxylated bridging carbon (C7) to uracil [41]. Cyanobacteria producing cylindrospermopsin are represented by filamentous species belonging mainly to the orders Nostocales and Oscillatoriales. To date, thirteen species have been reported as potent producers of this alkaloid. These species are believed to retain their current habitats and expand in the future [7]. Unlike the microcystin, which is predominantly intracellular, the CYL is predominantly extracellular due to the active release from the cyanobacterial cells [1,42].

CYL is known to inhibit chlorophyll production, photosynthesis, and phytoplankton growth [8,43,44]. On the other hand, Campos et al. reported no inhibitory effects of CYL on some marine and freshwater phytoplankton species when they are exposed to concentrations up to 179 µg/L [45]. Such concentrations of CYL (179 µg/L) were found to stimulate the growth of *Chlorella vulgaris*. On the other hand, the crude extract of *Aphanizomenon ovalisporum*, containing only 32 µg/L CYL, is highly toxic and significantly inhibits the growth of green algae [45]. At environmentally occurring concentrations (≤0.8 mg/L), CYL does not affect the growth of *Nannochloropsis* sp. and *Chlamydomonas reinhardtii*. Growth inhibition in these species was observed at a higher concentration of CYL (2.5 mg/L) [46]. A study by Bar-Yosef et al. revealed a possible allelopathic effect of CYL related to its effect on the alkaline phosphatase (ALP) activity in *Chlamydomonas* and *Debarya* sp. [47].

The biological and ecological role of the CYL, similarly to that of microcystins, has so far been rarely considered and poorly understood [48]. In many cases, it is difficult to demonstrate direct evidence of allelopathic interactions in natural phytoplankton communities. Therefore, it is important to characterize the allelopathic effects in controlled experiments in order to clarify the nature of the released substances and their mode of action on the target organisms. In addition, it will be interesting to know whether cyanotoxins use similar mechanisms and have the same effects on their coexisting autotrophic competitors in freshwater and marine basins, and whether they affect equally both motile and non-motile phytoplankton.

Since such studies are relatively few, as target autotrophic organisms we decided to use two freshwater green algae (*Chlamydomonas asymmetrica* Korshikov 1927, which is motile, and *Scenedesmus obtusiusculus* Chodat 1913, which is non-motile) and one green marine microalga growing in waters with high levels of salinity (*Dunaliella salina* [Dunal] Teodoresco 1905). Members of these genera often co-occur with toxin-producing cyanobacteria [49,50,51,52]. Thus, the present study aims to examine a current and still unresolved issue related to the effects of the cyanotoxins microcystin-LR and cylindrospermopsin on green algae (*Chlamydomonas asymmetrica*, *Dunaliella salina*, and *Scenedesmus obtusiusculus*) that frequently coexist with toxin-producing cyanobacteria, as well as to expand knowledge about the mechanisms by which these cyanotoxins act. Additionally, in the long term, to create an opportunity to predict the effect of the released cyanobacterial toxins on individual aquatic communities, and to take adequate preventive actions.

## 2. Results

### 2.1. Density and Morphology of the Algae Exposed to Cyanotoxins

The influence of MC-LR and CYL on the cell density and development of the investigated algal cultures after exposure for 15, 24, 48, and 72 h showed a significant inhibition of the algal growth (Table 1, Figure 1) and partially of their motility (Figure 2), compared to the controls.

The growth inhibition in all the green algae tested was observed after 24 h of exposure to cyanotoxins and ranged between 15.49 and 48.57% for MC-LR and between 11.97 and 42.86% for CYL (Table 1). After 72 h of exposure, growth inhibition reached 58.86–74.18% for MC-LR and 60–83.33% for CYL (Table 1). MC-LR had a more pronounced inhibitory effect on the motile green algae *Chlamydomonas asymmetrica*. The growth of the non-motile green alga *Scenedesmus obtusiusculus* was affected by both cyanotoxins, with the strongest effect (83.33%) found at 72 h of exposure with CYL (Table 1). Algal growth inhibition by both cyanotoxins was a time-dependent effect (Figure 1). After 15 h of treatment, a weak inhibitory effect was observed only under the influence of CYL on *Scenedesmus obtusiusculus*. With the increasing exposure time (24, 48, and 72 h), this effect occurred in all three green algae, increasing and becoming significant at 48 and 72 h.

The motility of the algae *Chlamydomonas asymmetrica* and *Dunaliella salina* was also affected by the cyanotoxins (Figure 2). The motility of the starting cultures (time 0) was 62% for *Chlamydomonas asymmetrica* and 80% for *Dunaliella salina*. After 15 h of exposure, in both algae the motility decreased, especially under the influence of MC-LR (14% motile cells in the culture of *Chlamydomonas asymmetrica* and 20% motile cells in the culture of *Dunaliella salina*). After 24 h, in the culture of *Chlamydomonas asymmetrica*, this initial effect was overcome and even turned into a motility-stimulating effect for the cells exposed to MC-LR (Figure 2A). *Dunaliella salina* reacted quite differently. A reduced number of motile cells was observed in *Dunaliella salina*, and this effect became more pronounced for CYL with the increasing exposure time (Figure 2B). In this alga, at 24 h, the initial motility inhibitory effect of MC-LR was also overcome, but at 48 and 72 h, a time-dependent motility inhibitory effect was observed (Figure 2B).

A microscopic analysis was carried out, taking into account the changes in the algal morphology. Morphological changes were associated with adverse effects on the algal cells. The most frequently observed morphological changes under the influence of cyanotoxins were changes in the shape of the cells and varying degrees of granulation of the cytoplasm (Figure 3). The investigated green algae responded differently to the exposure with cyanotoxins.

The control untreated algal culture of *Chlamydomonas asymmetrica* (Figure 3A) consisted of single cells with an intact protoplast. Morphological changes were not detected until 72 h, when only few palmelloid clusters of cells were found in places. This reaction is typical for representatives of the genus *Chlamydomonas* in conditions of biotic and abiotic stress [53]. In our case, palmelloids can be explained by a depletion of the nutrient medium.

After 15 h of CYL exposure, granulation of the protoplast was observed (Figure 3A). This effect persisted over time and was clearly visible after 48 and 72 h of exposure to CYL. In addition, after 48 h, *Chlamydomonas asymmetrica* responded to CYL with the formation of palmelloids (Figure 3A, CYL, 48 h). This change in the morphology is most likely due to the stress caused by exposure to the toxin. After 72 h of exposure to CYL, both single cells and palmelloid micro-colonies (Figure 3A, CYL, 72 h) were detected, and the single cells predominated.

Granulation of the protoplast was observed as early as 15 h after exposure to MC-LR (Figure 3A, MC-LR, 15 h). This effect on the morphology of *Chlamydomonas asymmetrica* was observed until the end of the MC-LR treatment (72 h). The MC-LR treatment induced cell crowding and palmelloid micro-colony formation similar to the effect of CYL with the difference that this effect appeared earlier (15 h), and was overcome as early as 48 h after exposure (data not shown).

Untreated cells of *Dunaliella salina* were single and characterized by an intact protoplast, a clearly distinguishable nucleus, a chloroplast, and a flagellum (Figure 3B). Both cyanotoxins caused the granulation of protoplasts and the loss of the flagella after 72 h of exposure (Figure 3B, CYL, MC-LR, 72 h). After 48 h of exposure to CYL, a change in the cell shape was also observed—in some cases, rounding occurred, and in others, there was a formation of slightly expanded areas of the cell (Figure 3B, CYL, 48 h), which was possibly caused by a change in the structure of the cell membrane [54].

The untreated algal culture of *Scenedesmus obtusiusculus* (Figure 3C) consisted of single cells and three- or four-celled coenobia. Single cells predominated. The protoplast was intact, with a clearly visible nucleus and a conspicuous chloroplast. Algae treated with MC-LR and CYL lost their coenobial structure, the cells became rounded, and the protoplast became granular. These changes in the morphology were observed 15 h after the cyanotoxins exposure and they persisted up to 72 h (Figure 3C).

### 2.2. Intracellular ATP in Algae Exposed to Cyanotoxins

The activity of the ATP synthase in algae exposed to cyanotoxins was analyzed by measuring the intracellular adenosine triphosphate (ATP) levels and comparing them with those of the control untreated algae. Reduced ATP production as a consequence of the effects of MC-LR and CYL was observed in all three algal cultures (Figure 4). However, each of them was affected to a different degree. The effect on *Dunaliella salina* culture was most pronounced. Both toxins significantly reduced ATP production in this alga (74.33% for CYL and 70.3% for MC-LR). The effect of CYL was slightly stronger compared to MC-LR, but there was no statistical significance.

The effect exerted by both toxins on *Chlamydomonas asymmetrica* was significantly weaker. The decrease in ATP production in this case was 14% compared to the control. The effects caused by both toxins were equal in strength.

The obtained data about the ATP levels are consistent with the effects of the studied cyanotoxins on the motility of *Chlamydomonas asymmetrica* and *Dunaliella salina*—in general, they were strongly affected in *Dunaliella salina*, and less affected or stimulated in *Chlamydomonas asymmetrica*.

In the *Scenedesmus obtusiusculus* culture, we found a significantly stronger effect on the ATP production by MC-LR (decrease of 45%) compared to the control (Figure 4C). The reduced ATP production in *Scenedesmus obtusiusculus* after exposure to CYL was 5% (Figure 4C). The effect exerted by MC-LR on *Scenedesmus obtusiusculus* was also significantly higher compared to CYL (*p* = 0.0495, non-parametric Mann–Whitney U-test).

Our results showed that the ATP synthase activity in algae exposed to cyanotoxins was also affected, resulting in decreased ATP levels.

### 2.3. Photosynthetic Activity of the Algae Exposed to Cyanotoxins

To study the photosynthetic activity, two separate experiments were carried out with the three strains of algae—*Chlamydomonas asymmetrica*, *Dunaliella salina*, and *Scenedesmus obtusiusculus*. Algae were exposed to MC-LR and CYL and changes in the chlorophyll fluorescence parameters were followed in dynamics (0, 15, 24, and 48 h). The effects of MC-LR and CYL treatments on the maximum quantum yield of PSII (F_v_/F_m_), on the non-photochemical quenching parameter (NPQ) reflecting downregulation of PSII as a protective mechanism against excess light, as well as on the quantum yield of non-regulated energy dissipation (Y(NO)) are presented in Figure 5, Figure 6 and Figure 7, respectively.

Analyses were performed based on a minimum of four replicates for each strain and each cyanotoxin. The effect of methanol administered alone as a control (nutrient medium containing 1% methanol) was also followed. Data showed that *Scenedesmus obtusiusculus* is characterized by the highest F_v_/F_m_ of PSII and *Dunaliella salina* by the lowest (Time 0). During the experiment, the F_v_/F_m_ values decreased from 0.46 to 0.33 in the control culture of *Chlamydomonas asymmetrica* (Figure 5A), while in the control culture of *Dunaliella salina*, a significant decrease was observed after 24 h (F_v_/F_m_ = 0.199) and 48 h (F_v_/F_m_ = 0.16) (Figure 5B). In *Scenedesmus obtusiusculus*, this parameter remained stable and did not change significantly during the experiment (Figure 5C). Overall, the treatment with methanol alone had a negative effect on the tested strains, which was most pronounced for *Chlamidomonas asymmetrica* and *Dunaliella salina*, while no effect of methanol was observed in *Scenedesmus obtusiusculus* during the experiment (Figure 5).

The strongest negative effect of MC-LR on F_v_/F_m_ was observed in *Chlamydomonas asymmetrica*. The maximal PSII quantum yield significantly decreased, reaching 40% of the control values after 15 h of exposure (Figure 5A; blue symbols). The response was weakest in *Scenedesmus obtusiusculus*, with the most sensitive changes observed at the end of treatment (48 h) (Figure 5C). No significant changes were observed in *Dunaliella salina* exposed to MC-LR compared to the control values at each time point (Figure 5B).

The effect of CYL was weaker than that of MC-LR in the motile algae strains. The inhibition of F_v_/F_m_ at 15 h and 24 h was about 20–25% in *Chlamydomonas asymmetrica* and *Scenedesmus obtusiusculus* (Figure 5A,C; red symbols). What is interesting to note is that F_v_/F_m_ was significantly higher in *Dunaliella salina* exposed to CYL after 15 h and it remained higher, despite the fact that the difference was not significant after 24 h of treatment (Figure 5B). A decrease in F_v_/F_m_ (compared to control) was recorded in *Dunaliella salina* only at 48 h.

We also analyzed the non-photochemical quenching of chlorophyll fluorescence (NPQ) (Figure 6). This parameter reflects the ability of chlorophyll-containing organisms to dissipate excess light energy via the xanthophyll cycle and, thereby, protect themselves from damage.

Treatment with methanol alone reduced NPQ over time in *Chlamydomonas asymmetrica* and *Scenedesmus obtusiusculus* (Figure 6A,C; black symbols). NPQ was strikingly lower in *Dunaliella salina* compared to other strains at time “0”, and remained lower and relatively stable over the course of the experiment (Figure 6B; black symbols).

In samples treated with MC-LR, NPQ was significantly inhibited over the experimental period in *Chlamydomonas asymmetrica* (Figure 6A). This parameter decreased by 65% compared to control samples. In *Scenedesmus obtusiusculus*, NPQ initially decreased by 16.7% at 15 h of treatment, but later (at 24 and 48 h) it became significantly higher than control values (Figure 6C). In *Dunaliella salina*, no significant changes in NPQ were observed during the experiment (Figure 6B). CYL affects NPQ in a similar manner as MC-LR in all studied algae strains (Figure 6).

The changes in the quantum yield of the uncontrolled energy dissipation Y(NO) in PSII are presented in Figure 7. An increase in this parameter indicates that both the photochemical energy conversion and the protective regulatory mechanisms are ineffective and that the algae cannot cope with the incident radiation.

It is interesting to note that the control algal culture of *Dunaliella salina* was characterized by significantly higher Y(NO) compared to the other two algae at time “0” (Figure 7B), and that this parameter remained significantly higher throughout the experiment. No substantial changes over the time were observed for *Chlamydomonas asymmetrica* and *Scenedesmus obtusiusculus* (Figure 7A,C). The methanol alone does not significantly change the quantum yield of the unregulated energy dissipation Y(NO) in PSII over the time in all three algae species (Figure 7; black symbols).

Exposure to MC-LR significantly increased, by 86%, the proportion of Y(NO) in *Chlamydomonas asymmetrica*, especially after 15 h (Figure 7A; blue symbols). In *Scenedesmus obtusiusculus*, there were no significant changes in the analyzed parameter. In *Dunaliella salina* there was some decrease in Y(NO) after 15 h (compared to the control), but at a later stage (48 h), the values of Y(NO) significantly increased (Figure 7B; blue symbols). In general, CYL had a similar effect on Y(NO) in all the three algal strains included in the study (Figure 7; red symbols). The most pronounced effect was observed in *Chlamydomonas asymmetrica* when Y(NO) picked the highest value at 15 h and remained higher over the experimental time (Figure 7A; red symbols).

## 3. Discussion

The eutrophication of the aquatic ecosystems and climate changes associated with global warming has increased the frequency and intensity of the cyanobacterial blooms and related cyanotoxins [9,19,20,55]. As a result of these changes, the ratio between the mainly represented taxonomic groups of algae and cyanobacteria, forming the phytoplankton in freshwater basins, has also changed in recent years. It has been reported that cyanobacteria are a dominant group in many water bodies [56,57,58,59,60,61]. In some water bodies, there is even a displacement of the Chlorophyta, which traditionally occupy the first place in the phytoplankton of freshwater basins, by representatives of cyanobacteria [57,60,62]. In all cases, however, the representatives of these groups of phytoplankton interact with each other and cyanobacteria always have an advantage. It is quite plausible that this is due to the second metabolites (including cyanotoxins) produced by the cyanobacteria, which reach a high concentration during the “blooms”. Despite the frequent identification of cyanotoxins such as microcystins (including MC-LR) and cylindrospermopsin (CYL) in water bodies, their effects on phytoplankton communities are poorly studied [8,9,44,63,64]. There are many questions related with the allelopathic effects of these secondary metabolites. Reports related to the effects of MC-LR on autotrophic organisms are limited. Microcystins have been found to induce a reduction in aquatic plant biomass, a reduction in protein phosphatases 1 and 2A, lipid peroxidation, oxidative stress, a decrease in photosynthetic activity, and even apoptosis in aquatic plant cells. [9,13].

In the present study, the cyanotoxins MC-LR and CYL (1 µg/mL) were found to inhibit the growth and development of green algae (*Chlamydomonas asymmetrica*, *Dunaliella salina*, and *Scenedesmus obtusiusculus*) over time (Figure 1). The inhibitory effects were most pronounced after 72 h of exposure in *Scenedesmus obtusiusculus*, followed by *Chlamydomonas asymmetrica* and *Dunaliella salina*. Similarly, *Aphanizomenon ovalisporum* crude extract containing 32 mg/L CYL was found to be highly toxic and significantly reduced the growth of the green alga *Chlorella vulgaris* [45]. Pinheiro et al. also reported a growth inhibition of the green algae *Chlorella vulgaris*, *Nannochloropsis* sp., and *Chlamydomonas reinhardtii* by using a crude extract containing CYL at concentrations higher than 2.5 mg/L [46]. On the other hand, when these authors used pure toxins (CYL and MC-LR) at concentrations 5 µg/L, 18.4 µg/L, and 179 µg/L, after three days of exposure, they observed stimulation of the algal growth [45]. However, the published data on the subject are quite diverse. This gave the researchers reason to assume that some green algae may have developed a defense mechanism against CYL by the induction of oxidative stress [7]. The species of green algae that we studied clearly do not belong to this group.

The role of MC-LR as an allelochemical is also disputed. Some authors reported an inhibitory effect exerted by the cyanotoxin on the motility and growth of green algae [25,65], others found no such effect [46], and thirdly, others found a stimulating effect [63]. The data in the present study clearly demonstrated an inhibitory effect on the growth and development of *Chlamydomonas asymmetrica*, *Dunaliella salina*, and *Scenedesmus obtusiusculus*, exerted by both types of cyanotoxins MC-LR and CYL. These results correlate with the measured maximum photochemical activity (F_v_/F_m_) (Figure 5). The maximum quantum efficiency values were reduced, thereby limiting the photosynthesis, which could explain the reduced growth. MC-LR had a significant negative effect on the F_v_/F_m_ of *Chlamydomonas asymmetrica* as early as 15 h after exposure to the cyanotoxin, and it was overcome at 48 h. In *Scenedesmus obtusiusculus* and *Dunaliella salina*, the response to the cyanotoxin was weaker with a peak at 48 h after exposure. Interestingly, CYL did not negatively affect F_v_/F_m_ in *Dunaliella salina*, as even more F_v_/F_m_ was significantly higher at 15 h compared to the non-toxin treated samples. A significant decrease in F_v_/F_m_ was registered only in *Dunaliella salina* at 48 h. This means that the reason for the inhibited growth of algal cultures under the influence of CYL must be sought elsewhere, and the mechanisms of action of these two cyanotoxins are different.

Most of the cyanobacterial inhibitory allelochemicals are directed against the oxygenic photosynthetic processes of other cyanobacteria or algae. Given this, we followed the change of two more of the chlorophyll fluorescence parameters, namely the non-photochemical quenching of chlorophyll fluorescence (NPQ) (Figure 6) and the quantum yield of the unregulated energy dissipation Y(NO) in PSII (Figure 7). The influence of MC-LR on *Dunaliella salina* led to an increase in NPQ after 15 h of exposure, after which its value significantly decreased (72 h). This could result in stress for the algae at the beginning of the exposure and an inability to overcome this effect at a later stage. In the other two algae, the NPQ values remained relatively stable for the period 15–48 h. The high values of the unregulated energy dissipation quantum yield Y(NO) in PSII of *Chlamydomonas asymmetrica* (Figure 7A) indicate that the photochemical energy absorption and protective regulated mechanisms are inefficient and that the alga has a serious problem regarding the absorption of light.

Our data on the photosynthetic activity of the green algae *Chlamydomonas asymmetrica*, *Dunaliella salina*, and *Scenedesmus obtusiusculus* after treatment with the cyanotoxins MC-LR and CYL lead to the conclusion that (1) the photosynthetic activity was significantly affected by MC-LR, with significant changes in all three parameters of the observed chlorophyll fluorescence and that (2) the photosynthetic activity of the green algae included in the study was less affected by CYL. This cyanotoxin changes the parameter non-photochemical quenching of chlorophyll fluorescence (NPQ) in *Dunaliella salina*, which decreased its values over the time (15–48 h), meaning that it acts by inducing stress in the treated algae.

Algal cultures responded to the cyanotoxins MC-LR and CYL with changes in their morphology expressed by changes in their cell shape, the granulation of the protoplast, the loss of flagella of the motile forms, the formation of palmeloid micro-colonies (in *Chlamydomonas asymmetrica*), or the loss of the coenobial form (in *Scenedesmus obtusiusculus*). Similar effects on green algal morphology have been observed by other researchers following exposure to micropollutants (copper, cadmium, perfluoroethanesulfonic acid, and paraquat) [66].

Microcystins are known to inhibit protein phosphatases in plant and animal cells and are thus potentially capable of acting as allelochemicals [6]. In animal cells, MC-LR leads to an increase in reactive free radicals (ROS), which determine mitochondrial permeability [67]. Mitochondria have been suggested to play a major role in MC-LR-induced apoptosis, suggesting two possible pathways—a decrease in the mitochondrial membrane potential and an increase in the permeability of the outer mitochondrial membrane [68]. The results of our study indicated that the mitochondria are target organelles for MC-LR in *Chlamydomonas asymmetrica*, *Dunaliella salina*, and *Scenedesmus obtusiusculus* cells as well. This was indicated by the reduced production of ATP (Figure 4), as a consequence of the impact of MC-LR and CYL in all three algal cultures, as well as the reduced motility of *Dunaliella salina* and *Chlamydomonas asymmetrica*. However, all studied algal cultures were affected by MC-LR and CYL to a different degree. The motility inhibition did not paralyze the algae as previously reported by other researchers [25]. They demonstrated that MC-LR (10 ng/mL) is able to reduce the motility of the green alga *Chlamydomonas reinhardtii* from 100% to 10% [25].

Our results indicate that MC-LR and CYL affect the growth rate, motility, ATP production, and photosynthesis of the green algae *Chlamydomonas asymmetrica*, *Dunaliella salina*, and *Scenedesmus obtusiusculus*. The allelopathic potential of cyanotoxins on competing photosynthetic organisms in aquatic communities is more complex, and environmental factors must also be considered to understand its significance.

## 4. Materials and Methods

### 4.1. Algae, Culture Conditions and Exposure to Cyanotoxins

Three green clonal axenic algae (*Chlamydomonas asymmetrica* PACC 5120, *Dunaliella salina* PACC 8619, and *Scenedesmus obtusiusculus* PACC 8853) kept in the Plovdiv Algal Culture Collection (PACC) at the Paisii Hilendatski University of Plovdiv were used in this study (Figure 8).

Intensive cultivation of the strains was carried out in a cultivation block, described by Dilov et al. [69] at 26 °C with a light/dark cycle of 15/9 h (light intensity of 10–20 μmol photon s^−1^ m^−2^, provided by cool-white, fluorescent tubes). *Chlamydomonas asymmetrica* and *Scenedesmus obtusiusculus* were grown in BBM1 (Bold’s Basal medium) [70], while the culture of *Dunaliella salina* was grown in Eddy’s medium [71]. The initial cell density was controlled at the beginning of the light period by diluting with nutrient medium to a concentration of 2.5 × 10^5^ cells/mL.

In the exponential growth phase, the algal strains were transferred in 6-well culture plates (TPP, Trasadingen, Switzerland) in triplicates (1 × 10^6^ cells/mL in a total volume of 5 mL), and were exposed to the cyanotoxins microcystin-LR and cylindrospermopsin (Figure 9) at a concentration of 1 µg/mL for 15, 24, 48, and 72 h. This concentration was chosen based on similar publications in the field (to be comparable) and the WHO guidelines, where the reference values for MC-LR and CYL in drinking water are 1 µg/L and 0.7 µg/L, respectively, but in aquatic environments and cyanobacterial blooms, these toxins are naturally occurring at concentrations of 1–100 mg/L [72]. Microcystin-LR (MC-LR) and cylindrospermopsin (CYL) were purchased from Merck KGaA (Darmstadt, Germany). Toxins were dissolved in the respective nutrient medium. For the experiment examining photosynthetic activity, the toxins were dissolved in methanol and the final concentration of methanol in the samples was 1%. For this experiment, the control also contained 1% methanol.

### 4.2. Counting of Cell Density and Morphological Analysis of the Algae

A Bürker’s chamber was used to determine the effect of the cyanotoxins on the cell density and development of the tested algal cultures after exposure for 15, 24, 48, and 72 h. Additionally, the density of the algal cultures was measured at the indicated times with a NanoDrop 2000 UV–Vis spectrophotometer (Thermo Fisher Scientific, Wilmington, DE, USA) at 600 nm wavelength.

Changes in the motility of *Chlamydomonas asymmetrica* and *Dunaliella salina* were also recorded using a Bürker’s chamber by counting the number of motile and non-motile cells in the samples. The morphological analysis of the algal cells was performed by a standard Magnum-T light microscope (Medline Scientific, Chalgrove, Oxon, UK) equipped with a high definition digital camera Si-3000 and XLiCap software (Medline Scientific, Chalgrove, Oxon, UK). Observed changes in the algal morphology were photo-documented.

### 4.3. Measurement of Intracellular Adenosine Triphosphate (ATP) Concentration

The levels of synthesized ATP after exposure of the green algae to cyanotoxins were measured using a luminescent assay. After exposure of the algal cultures to the cyanotoxins MC-LR and CYL for 15 h, the cells were washed with sterile phosphate-buffered saline (PBS) and lysed with liquid nitrogen. The ATP concentration was measured using a standard ATP Determination Kit (A22066, Molecular Probes Inc., Eugene, OR, USA) according to the manufacturer’s protocol. The assay was based on a bioluminescent reaction between D-luciferin and ATP catalyzed by the enzyme luciferase [73]. Sensitivity of the assay allows detection of ATP in a concentration of up to 0.1 picomole. Bioluminescence measurements were performed using a GloMax^®^ 20/20 luminometer (Promega Corporation, Madison, WI, USA).

### 4.4. Measurement of Photosynthetic Activity of the Algae

Photochemical efficiency of photosynthesis was analyzed by IMAGING-PAM fluorometer (MAXI version; Heinz Walz GmbH, Effeltrich, Germany). Samples were dark adapted for 30 min prior to the determination of minimum (F_0_) and maximum (F_m_) fluorescence. The maximum quantum yield of PSII photochemistry (F_v_/F_m_) was determined as (F_m_ − F_0_)/F_m_. During sample adaptation to the specific light intensity (56 μmol m^−2^ s^−1^), saturating pulses were applied in order to determine the steady-state fluorescence (F’) and the maximum fluorescence (F_m_’) in the light. The non-photochemical dissipation of absorbed light energy (NPQ) was determined using the equation NPQ = (Fm − F_m_’)/F_m_’) [74].

### 4.5. Statistical Analysis

Chlorophyll fluorescence data are presented as means ± SE from two independent series of experiments (4 measurements each and for each algae strain). The significant differences are determined by the Student’s *t*-test. All other experiments were conducted in triplicates and results are presented as mean ± SD. Statistical significance of the results was determined by the non-parametric Mann–Whitney U-test using the StatView software (SAS Institute Inc., Cary, NC, USA). The differences were compared with the corresponding controls, if not otherwise stated. Data were considered significant when *p* < 0.05.

## 5. Conclusions

Through a combination of several research methods, the allelopathic potential of the cyanotoxins microcystin-LR and cylindrospermopsin on green algae (*Chlamydomonas asymmetrica*, *Dunaliella salina*, and *Scenedesmus obtusiusculus*) was established. The time dependence of these effects on the cell density, growth, development, morphology, and photosynthetic activity of the exposed algae was monitored. Changes in the morphology and motility that could serve to identify these species as bioindicators for presence of cyanotoxins in water bodies were described. The data obtained during the study complement and expand the knowledge about the mechanisms of action of the cyanotoxins. The results obtained in the course of the study reveal the relationship between toxin-producing cyanobacteria and the possible allelopathic influence of cyanotoxins on the examined green algae.

## Figures and Tables

**Figure 1 plants-12-01403-f001:**
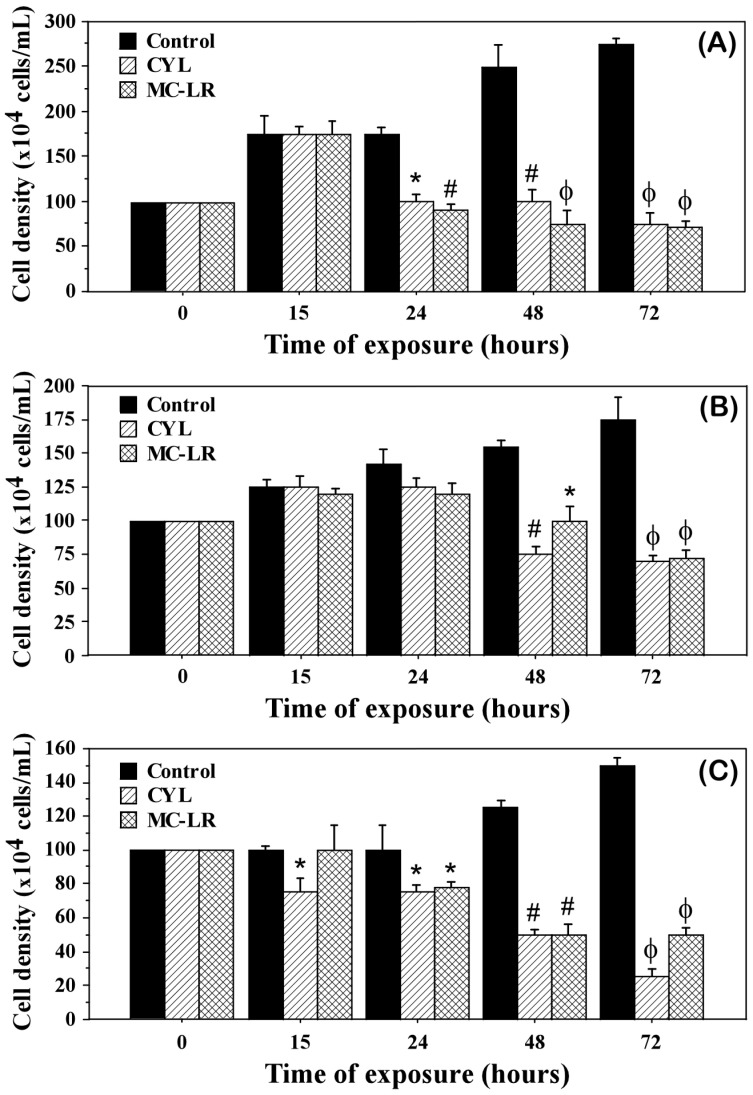
Cell density of the green microalgae *Chlamydomonas asymmetrica* (**A**), *Dunaliella salina* (**B**), and *Scenedesmus obtusiusculus* (**C**) exposed to 1 µg/mL of microcystin-LR (MC-LR) or cylindrospermopsin (CYL) for 15, 24, 48 and 72 h. Control: non-treated cell cultures. Results are presented as means ± SD. * *p* < 0.05, # *p* < 0.01, ϕ *p* < 0.001, as determined by Mann–Whitney U test vs. control.

**Figure 2 plants-12-01403-f002:**
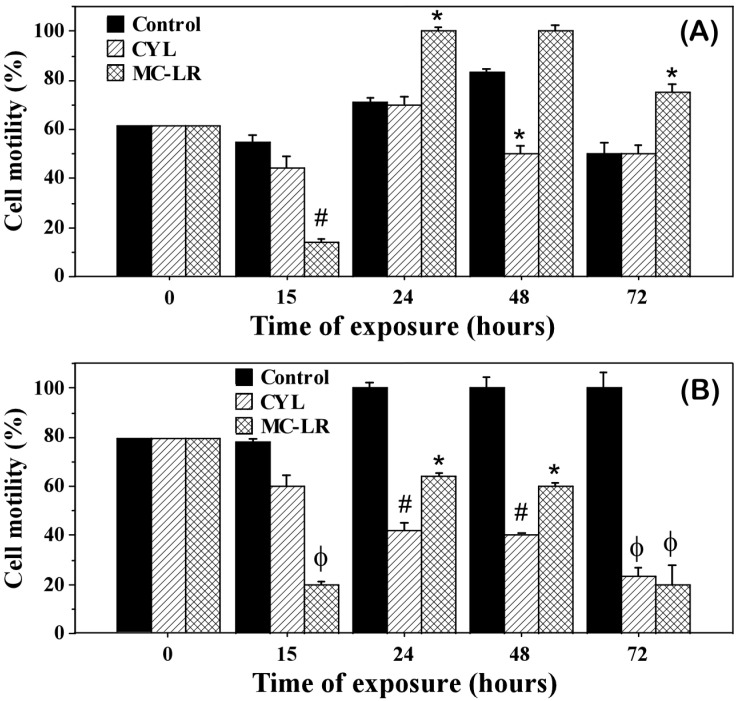
Motility of the green microalgae *Chlamydomonas asymmetrica* (**A**) and *Dunaliella salina* (**B**), after exposure to 1 µg/mL of microcystin-LR (MC-LR) or cylindrospermopsin (CYL) for 15, 24, 48, and 72 h. Control: non-treated cell cultures. Results are presented as means ± SD. * *p* < 0.05, # *p* < 0.01, ϕ *p* < 0.001, as determined by Mann–Whitney U test vs. control.

**Figure 3 plants-12-01403-f003:**
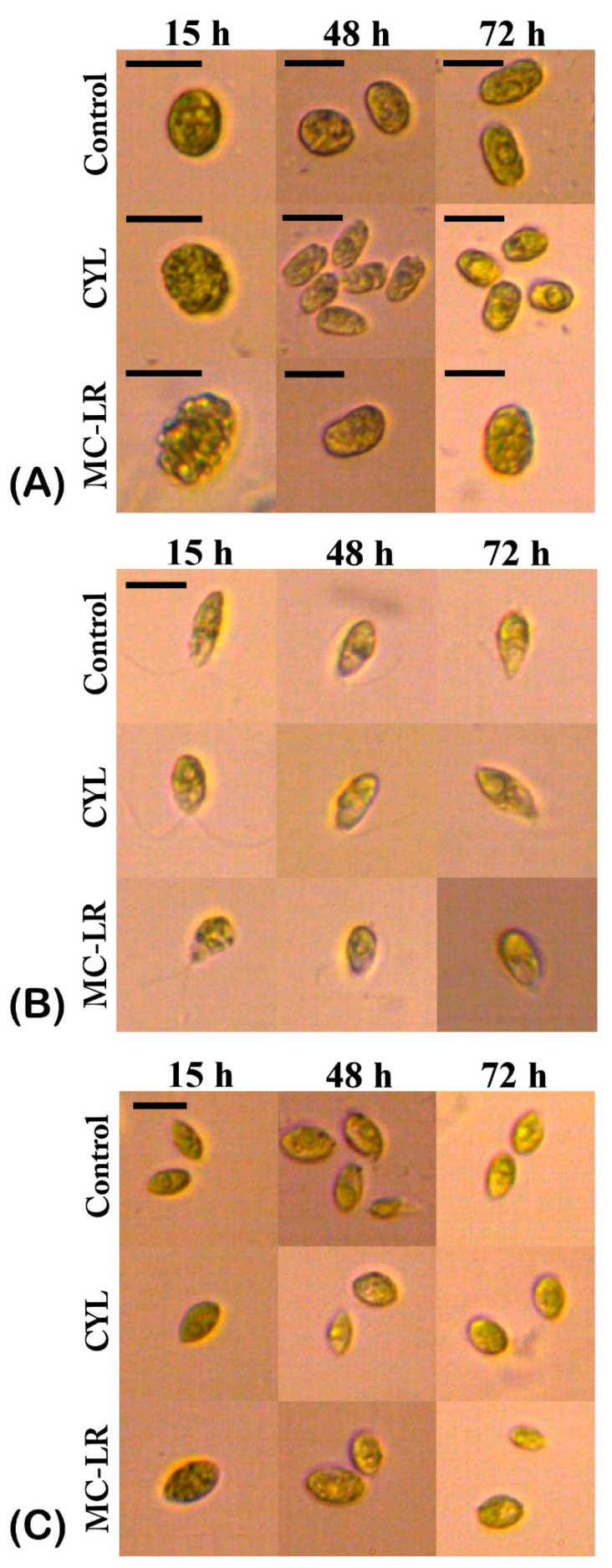
Morphology of *Chlamydomonas asymmetrica* (**A**), *Dunaliella salina* (**B**), and *Scenedesmus obtusiusculus* (**C**), after exposure to 1 µg/mL MC-LR and CYL for 15, 24, 48, and 72 h. Control: non-treated cell cultures. Bar = 10 µm.

**Figure 4 plants-12-01403-f004:**
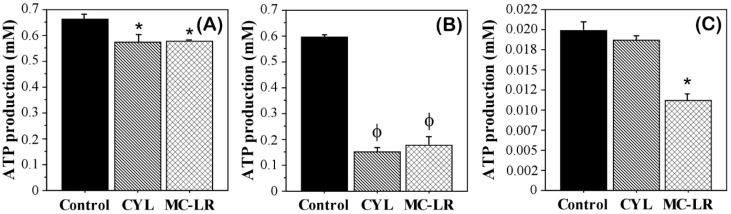
Influence of microcystin-LR (MC-LR) and cylindrospermopsin (CYL) on the production of adenosine triphosphate (ATP) in *Chlamydomonas asymmetrica* (**A**), *Dunaliella salina* (**B**), and *Scenedesmus obtusiusculus* (**C**) after exposure for 15 h. Control: non-treated cell cultures. Results are presented as means ± SD. * *p* < 0.05, ϕ *p* < 0.001, as determined by Mann–Whitney U test vs. control.

**Figure 5 plants-12-01403-f005:**
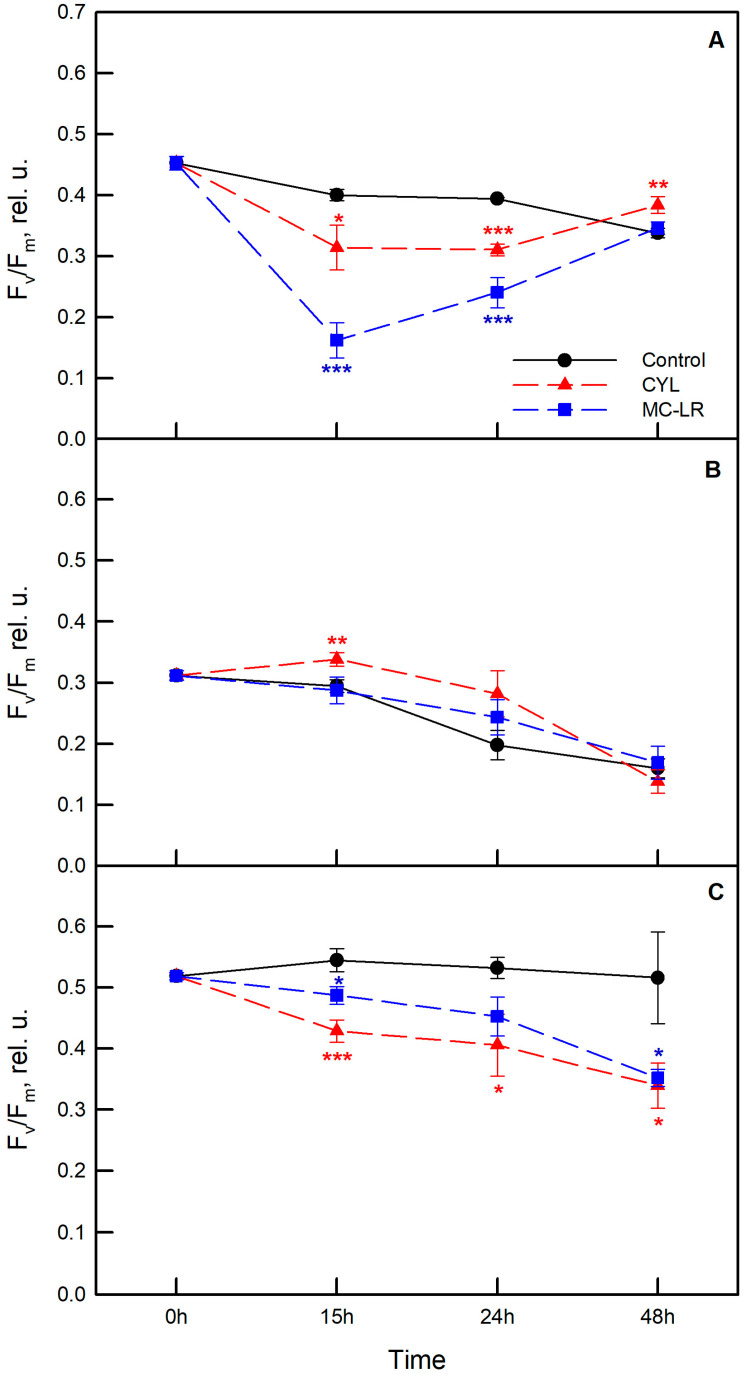
Effect of microcystin-LR (MC-LR; blue symbols) and cylindrospermopsin (CYL; red symbols) on the maximum photochemical activity of photosystem II (F_v_/F_m_) in dark-adapted algae strains *Chlamydomonas asymmetrica* (**A**), *Dunaliella salina* (**B**), and *Scenedesmus obtusiusculus* (**C**) after 15, 24, and 48 h of exposure to the cyanotoxins. Results are presented as means ± SE. Asterisks (* *p* < 0.1; ** *p* < 0.05; *** *p* < 0.01) indicate statistically significant difference with non-toxin treated cell cultures (control—nutrient medium containing 1% methanol).

**Figure 6 plants-12-01403-f006:**
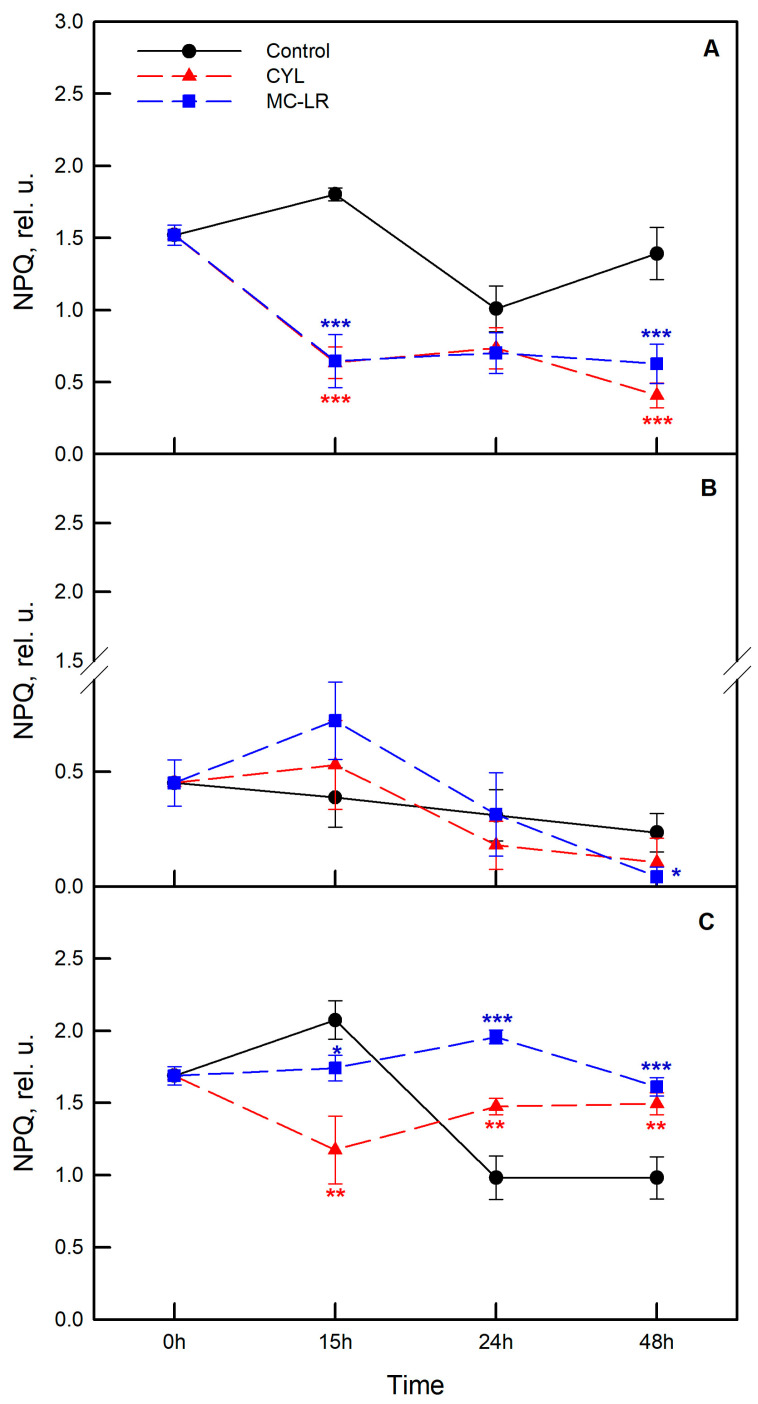
Effect of microcystin-LR (MC-LR; blue symbols) and cylindrospermopsin (CYL; red symbols) on the non-photochemical quenching of chlorophyll fluorescence (NPQ) in *Chlamydomonas asymmetrica* (**A**), *Dunaliella salina* (**B**), and *Scenedesmus obtusiusculus* (**C**) after 15, 24, and 48 h of exposure to the cyanotoxins. Results are presented as means ± SE. Asterisks (* *p* < 0.1; ** *p* < 0.05; *** *p* < 0.01) indicate statistically significant difference with non-toxin treated cell cultures (control—nutrient medium containing 1% methanol).

**Figure 7 plants-12-01403-f007:**
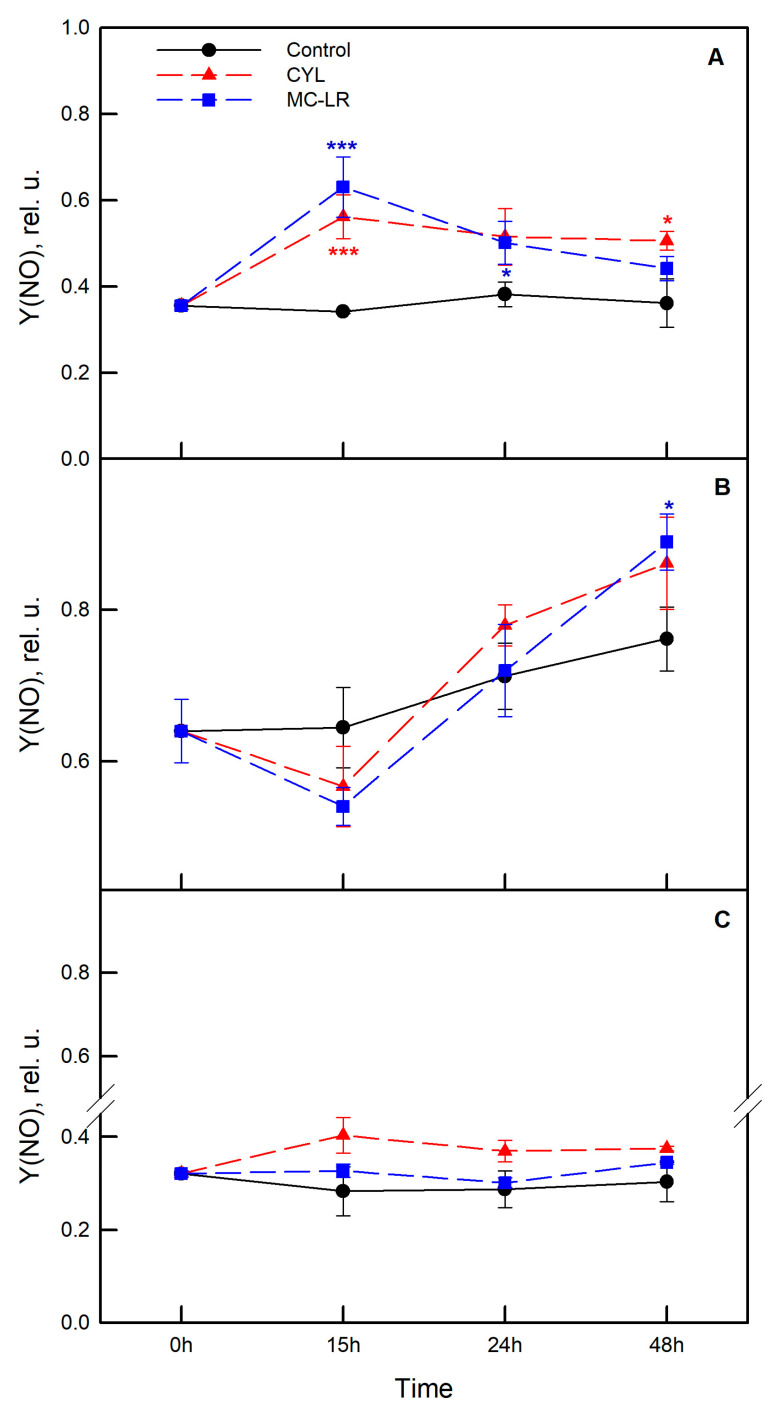
Effect of microcystin-LR (MC-LR; blue symbols) and cylindrospermopsin (CYL; red symbols) on the quantum yield of the uncontrolled energy dissipation Y(NO) in *Chlamydomonas asymmetrica* (**A**), *Dunaliella salina* (**B**), and *Scenedesmus obtusiusculus* (**C**) after 15, 24, and 48 h of exposure to the cyanotoxins. Results are presented as means ± SE. Asterisks (* *p* < 0.1; *** *p* < 0.01) indicate statistically significant difference with non-toxin treated cell cultures (control—nutrient medium containing 1% methanol).

**Figure 8 plants-12-01403-f008:**
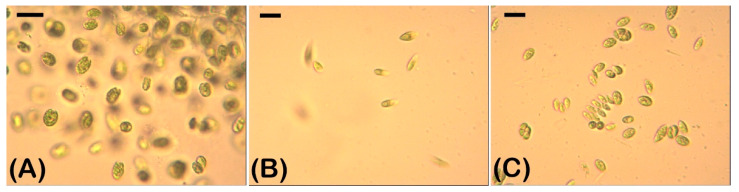
Algal strains used in the experiments. (**A**) *Chlamydomonas asymmetrica* PACC 5120, (**B**) *Dunaliella salina* PACC 8619, and (**C**) *Scenedesmus obtusiusculus* PACC 8853. Bar = 10 µm.

**Figure 9 plants-12-01403-f009:**
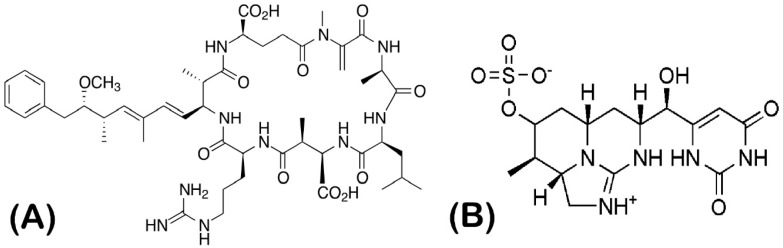
Chemical structure of the cyanotoxins used in the experiment. (**A**) Microcystin-LR (MC-LR) and (**B**) cylindrospermopsin (CYL).

**Table 1 plants-12-01403-t001:** Growth inhibition of *Chlamydomonas asymmetrica*, *Dunaliella salina*, and *Scenedesmus obtusiusculus* after exposure to MC-LR and CYL for 15, 24, 48, and 72 h.

Algal Species	Microcystin-LR (MC-LR) ^§^				Cylindrospermopsin (CYL) ^§^			
	15 h	24 h	48 h	72 h	15 h	24 h	48 h	72 h
*C. asymmetrica*	−	48.57 ^#^	70 ^ϕ^	74.18 ^ϕ^	−	42.86 *	60 ^#^	72.73 ^ϕ^
*D. salina*	4	15.49	35.48 *	58.86 ^ϕ^	−	11.97	51.61 ^#^	60 ^ϕ^
*S. obtusiusculus*	−	22 *	60 ^#^	66.67 ^ϕ^	25 *	25 *	60 ^#^	83.33 ^ϕ^

^§^ Data represent percentage (%) of inhibition compared to the control. (−) not determined (no inhibition). * *p* < 0.05, ^#^
*p* < 0.01, ^ϕ^
*p* < 0.001, as determined by Mann–Whitney U test vs. control.

## Data Availability

Data are contained within the article or available from the corresponding author upon request.

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
