# Peer review of "Allelopathic Potential of the Cyanotoxins Microcystin-LR and Cylindrospermopsin on Green Algae"

_plants, 2023, doi:10.3390/plants12061403_

Round 1
Reviewer 1 Report
In this study, the authors test the acute effects of 2 cyanotoxins (microcystin and cylindrospermopsin) on 3 species of microalgae following morphology, growth and photosynthetic parameters. I have questions about the relevance of this study based on short-term exposures of 3 species (represented by one strain each) and using the toxins in a purified form. In my opinion, this situation is very far from an allelopathic interaction, a theme emphasized by the authors as the motivation for the study.
title
the title would look better in a review on this subject, here it should be more specific in relation to the results obtained
abstract
should describe results more precisely:
inhibitory effects were detected.
affect photosynthesis to varying degrees
are vague statements
a conclusion is lacking
introduction
As commented, the authors focus on allelopathy but the study does not involve interaction between species in a scenario that mimics the natural environment.
The introduction should relate to the experiments presented and focus on the effects of cyanotoxins on the physiology of microalgae only
33 inhabitants of the most water bodies
please rephrase
34 many of them are able to cause damage with ...
please describe which kind of damage
include reference
37 The cytotoxic effects of these compounds have been well studied in animals and humans
please include reference
39 still no consensus on the ecological role of the cyanobacterial toxins
[1] please use a more appropriate reference, a review, a more recent study, related to cyanotoxins in general
40 lack of information on the effect of cyanobacterial "blooms" on individual plant
communities in the aquatic ecosystems, allelopathic influence of cyanotoxins on other groups of algae.
effect of cyanobacterial "blooms" or effect of cyanobacterial toxins?
effect on aquatic plants or allelopathic effect on algae?
why other groups of algae - since cyanobacteria is not a group of algae?
please specify your question
43 Cyanotoxins are thought to have allelopathic potential on other aquatic organisms
but references [2,3] are for microcystin only
45 The allelopathic interaction is widespread.
please delete too vague
paragraphs 45-63
can be more specific and organized
please begin defining allelopathy, and allelochemicals
cite a more general reference, a book or review, focus on aquatic environments, cyanobacteria and allelopathy
then specify the possible role of cyanotoxins as allelochemicals (with references), since other secondary metabolites can act as allelochemicals and this is mixed in the present version of the text
possible targets and mode of action
61 lack of such inhibitory effects and the role of microcystins as allelochemicals is disputed [12]
Reference 12 cite only microcystin, how it can be used to argue about the role of cyanotoxins as a whole?
the authors focus on microcystin in the following paragraphs, use this information later
74 before citing microcystin the authors should briefly describe this toxin
the same applies to cylindrospermopsin
75 and phytoplankton [11,18,19]
ref 19 incomplete
the authors previously cited other studies relating microcystin and phytoplankton, can be included here
79-82
this paragraph is lost can be incorporated in the previous one
ref 16 is incomplete
97 At ecologically significant concentrations, CYL does not affect the growth of
Nannochloropsis sp. and Chlamydomonas reinhardtii. Growth inhibition in these species was
observed only at concentrations of the CYL up to 2.5 mg/L
unclear, if the concentration cited (2.5 mg/L) is " ecologically significant"
102 The biological and ecological role of cyanotoxins...
but the authors should focus on their motivation: physiological effects on algae
109 most commonly detected cyanotoxins (microcystins and cylindrospermopsin)
the justification for the study of these two toxins must appear before their description
108 issue related to the effects of the most commonly detected cyanotoxins on other groups of algae
again cyanobacteria are not a group of algae
please specify the aim of this study
114 to predict the effect of the cyanobacterial blooms on individual plant communities in aquatic ecosystems
effect of the presence of cyanotoxins (?) not all blooms are toxic and the study will focus on MC and CYL
will these results help to predict the effect of cyanobacterial blooms?
Results
results should be described more carefully and descriptions can be more concise
figure 1 the authors cite in methods that they exposed a cell preparation of 1 x 10 (5) cells/mL to MC and CYL
however the figure does not show time 0
after 15 hours there were 10(6) cells/mL? how fast do these algae divide in this culture condition? observing the control, these cells grow relatively slowly
133 Scenedesmus obtusiusculus is the most sensitive to the effects of cyanotoxins according to this indicator, and Dunaliella salina the least.
did the authors compared these values (between species)? I understand that the statistics was performed for each toxin x control in fig 1
could the authors calculate an inhibition index? or percent?
134 The effect on algal cultures of CYL was more pronounced compared to the effect of MC-LR
did the authors compare these values (CYL x MC)? the statistics was performed for each toxin x control in fig 1
136 why motility of the algae Scenedesmus was not tested? is this species not motile? please explain
137 A reduced number of motile cells was observed in Dunaliella salina, where this effect for both toxins (MC-LR and CYL) became more pronounced with increasing exposure time.
for MC inhibition in relation to control seems greater at 15 h and 72 h
139 At the 15th hour of treatment, in both algae motility was significantly decreased, especially under the influence of MC-LR (14% and 20%, respectively)
14 and 20% of the value observed for the control or a reduction of 14 and 20%?
respectively?
142 Chlamydomonas asymmetrica it even turned into a motility-stimulating effect.
for MC only
143 initial motility inhibitory effect was also overcome, but at 48 and 72 h, a time-dependent motility inhibitory effect was observed by both cyanotoxins (Figure 2B)
inhibition observed at 15 h persisted for CYL and for MC
if the the magnitude of inhibition changed, this is not evident from fig 1
The effect exerted by CYL was more pronounced (Figure 2).
compare to?
fig 2 the figure does not show time 0
152-188 the changes described in the control condition and treatments can not be observed from fig 3
a larger number of cells should be presented, representative fields, better resolution
160 clusters of cells were found in places (?)
depletion of the nutrient medium.
isn't it a rich medium? is it expected that it becomes depleted after 72 h of culture?
195 The effect of CYL is slightly stronger compared to MC-LR.
is it significant?
204 The decrease in ATP production in this case was one-fold compared to the control
(?) one fold
207 The obtained data about the ATP levels completely correlate with the effects of the studied cyanotoxins on the motility
fig 2 motility 15 h both algae affected by MC
this is discussion
213 mitochondrial activity in algae exposed to cyanotoxins was also affected, resulting in decreased ATP levels
can the authors state that the cause of ATP reduction was mitochondrial activity?
225 The effect of methanol administered alone as a control
why methanol as control? this is not explained in methods either
234-241 Scenedesmus obtusiusculus is characterized by the highest Fv/Fm of PSII and Dunaliella salina with the lowest
the aim of these experiment was to compare treatment x control, not species (same for 261 and 282)
248 The effect of CYL was weaker than that of MC-LR in studied algae strains
even for Scenedesmus?
256 biological objects (?)
259 Treatment with methanol alone significantly reduced NPQ in Chlamydomonas
compared to which condition?
In Scenedesmus obtusiusculus NPQ (fig 6C)
281 the biological samples have serious problems to cope with the incident radiation.
please rephrase
286 Scenedesmus obtusiusculus (Figure 7A,B). 7C
286 The methanol alone does not significantly change the quantum yield
in comparison to which condition? over time?
297 In Dunaliella salina there was some decrease in Y(NO) at 15 h..
according to fig 7B only change after 48h
discussion
the text in this section needs to be improved to writing with scientific rigor, more concise and informative
effect on the growth and development - which parameters indicated effects on development?
307 the ratio between the mainly represented taxonomic groups of algae, forming the phytoplankton in freshwater basins, has also changed in recent years
reference? worldwide?
314 and something gives an advantage to the cyanobacteria
rephrase, be more specific, include references, it applies to the whole paragraph
318 effects on phytoplankton communities are poorly studied
ref 34 please substitute for published studies
316-323 the authors mix effects on plants and effects on phytoplankton
324-335 contradictory or diverse findings
please consider that some studies used cell extracts while other used purified cyanotoxins, the target species also varied, as well as toxin doses and time of exposure
This gave the researchers reason to assume that some green algae may have developed a defense mechanism against CYL
thus, these findings can not be interpreted as the existence of defense mechanisms
there are many reasons for the different outcomes
338 inhibitory effect exerted by the cyanotoxin on green algae
reference?
343 Maximum quantum efficiency values were reduced, thereby limiting the amount of organic substrate produced during photosynthesis, which could explain the reduced growth
growth inhibition could be the result of other physiological changes
it is not possible to draw such a direct relation from the results
359-375
the authors do not need to recapitulate the results, they should summarize all the photosynthetic effects of the toxins as done in 377
389 Hepatotoxins
MCs?
397 . This was also confirmed
also? this is the only indication
discussion lacks final paragraph integrating the results and their relevance
Methods
418 The initial cell density
for which experiment? Is this the inoculum? how did the authors determine the cell density?
422 10 x 10(4) cells/mL = 10(5) cells/mL
424 how did the authors dilute the toxins, in water? fresh dilutions?
conclusions
the allelopathic influence of the cyanotoxins microcystin-LR and cylindrospermopsin on green algae was established
as mentioned before, the study design does not mimic an allelochemical interaction (same for 479-480)
Changes in ultrastructure were not demonstrated
could serve to identify these species as bioindicators for presence of cyanotoxins in water bodies
how? this was not the purpose of the study
Reviewer 2 Report
The Allelopathic Potential of the Cyanotoxins on Green Algae has received a lot of interest in recent years, as seen by the abundance of studies on the topic. The primary emphasis of this manuscript was on the response of green algae (Chlamydomonas asymmetrica, Dunaliella salina, Scenedesmus Obtusiusculus) to Microcystin-LR and Cylindrospermopsin. Also, investigate the inhibitory effects and possible mechanism by studying the morphology change, cell density, grow and motility. It is easy to understand both the results and the conclusion. On the other hand, it was not clear where the originality was.
. However, it needs considerable improvements before considering for publication.
Although in the introduction, the authors did no clearly explain the novelty of this study. Why do you conduct this study again? The authors must clearly manifest the novelty and significance of this study.
For the methodology:
Could you please add more information about the strains used. Are they axenic strains or not?
Why to use only the concentration 1 µg/mL of cyanotoxins?
Try to update the refrences list by new bibliography citations
Reviewer 3 Report
In the current study by Teneva and colleagues, the allelopathic effects of microcystins and cylindrospermopsin on three algal species are investigated. The authors highlight the conflicting state-of-the-art of the field and aim to clarify the impact of cyanobacterial allelochemicals. The authors conduct several experiments to determine the effects of allelochemicals on cell density, motility, ATP concentration, morphology, and photosynthetic ability. The experiments are clear to interpret although comparisons are made that do not have statistical support. Please address the comments below:
Introduction:
- - please justify the use of the specific algal species
- - as little is known of the effects of cyanobacterial toxins on phytoplankton, please include a discussion on the effects of plankton-derived toxins
Line 122: There is no statistical proof that CYL affects these species more than MC-LR – that would require post-hoc tests. Either remove or prove statistically. These types of comparisons are made throughout the manuscript and should be addressed in all places.
Line 424: Is 1 ug/mL an ecology relevant concentration? Please cite environmental studies.
Round 2
Reviewer 1 Report
abstract
16 (microcystins, nodularins, anatoxins, saxitoxins and 16 cylindrospermopsin)
can be removed
By using a combination of different methods
can be removed
introduction
Cyanobacteria are prokaryotic organisms that can be found in most water bodies
REF lacking
do the majority of water bodies have cyanobacteria?
This is mainly 38 due to their secondary metabolites (cyanotoxins)
a group of secondary metabolites known as cyanotoxins.
Allelopathy is a 50 biological phenomenon...community
I suggest using these sentences in the beginning of this paragraph
it is important to know the factors determining the relationship: cyanotoxins – allelopathic effects – affected organisms. 58
it is important to determine the possible allelopathic effects of cyanotoxins and the affected organisms. 58
Allelochemicals are a variety of bioactive secondary metabolites that are not required 59 for the metabolism of the allelopathic organism
not essential, secondary metabolites?
Growth inhibition in these species was observed at concentrations of the CYL 110 up to 2.5 mg/L
This is unclear after the previous statement, please indicate the minimum concentration that caused inhibition
The biological and ecological role of the CYL, similarly to microcystins, nodularins, 114 saxitoxins and anatoxins, has so far been rarely considered
I suggest that the authors restrict citations to the studied toxins - CYL and MC
in freshwater and marine basins and 121
marine microalga growing in waters with high levels of salinity (Dunaliella salina
Does this marine species co exist with cyanobacteria that produces CYL or MC?
results
Growth inhibition ranged between 4-74.18% for MC-LR and 141 between 11.97-83.33% for CYL (Table 1
please give range of inhibition considering the same time point, ex ranged between x-y% for MC-LR at 24h and x-y% after 72 h
Morphological changes were not 185 detected until 72 h, when clusters of cells were found in places. This reaction could be 186
explained by depletion of the nutrient medium
Unclear, these clusters of cells are expected at control conditions?
Unclear, morphological changes were not detected until 72 h then the authors say that at 15 h of CYL exposure, granulation 187
of the protoplast was observed (Figure 3A) and this effect persisted over time and was clearly 188 visible after 48
both single cells and palmeloid micro-colonies were detected, 192 where the single cells were predominated. (data not shown)
MC-LR treatment induced cell crowding and palmelloid 200 micro-colony formation similar to the effect of CYL with the difference that this effect 201 appeared earlier (15 h), and was overcome as early as 48 h after exposure. (data not shown)
It would be interesting to show the above mentioned aspects in the figure
Both 204 cyanotoxins caused granulation of protoplasts and loss of flagella.
at which time? 48h?
a change in the cell shape was also observed – in some cases rounding and in 206 others formation of slightly expanded areas of the cell,
in fig 3B or data not shown?
209-214 changes in morphology are not evident in the figures
The 221 effect of CYL was slightly stronger compared to MC-LR, but without statistical 222 significance.
unnecessary
229-233
please describe decrease in both algae in % compared to control
The obtained data about the ATP levels completely correlate with the effects of the 234
Intracellular ATP levels correlated with the effects
In Scenedesmus obtusiusculus culture, we found a significantly stronger effect on ATP 238
production by MC-LR compared to the effect exerted by CYL (Figure 4C).
fig 4 caption : as determined by Mann–Whitney U test vs control.
In fig 4 * refers to MC x control, a comparison between MC and CYL was not tested
a tendency towards a decrease in Fv/Fm was observed in the 263 control culture of Chlamydomonas
Fv/Fm values decreased from X to Y in the 263 control culture of Chlamydomonas
The 277 inhibition of Fv/Fm was about by 20-25% in
which time point?
A decrease in Fv/Fm was 281
compared to control or to previous time points?
Treatment with methanol alone significantly reduced NPQ over the time in 287
in fig 6 * denote differences control x test
did the authors test (statistic analysis) the differences between time points in the same condition? (over time)
In Scenedesmus obtusiusculus NPQ initially decreased, 294 by 43% at 15 h of treatment
please check figure, if the value decreased 43% of the control in 15 h 6C
discussion
355 In the present study, the cyanotoxins MC-LR and CYL
please include concentration used and justify
see comment in methods
365 at concentrations higher than 5 µg/L
please specify concentration tested
371 inhibitory effect exerted by the cyanotoxin on green algae
on growth?
thereby limiting 376 the amount of organic substrate produced during photosynthesis,
limiting photosynthesis
Maximum quantum efficiency values were reduced, thereby limiting 376 photosynthesis
which could explain 377 the reduced growth.
386 mechanisms of action of these two cyanotoxins are different.
unclear, did the authors mean in terms of effects on PSII? both toxins affected photosynthesis
In the other two algae, 394 NPQ remained lower
Decreased in relation to control? decreased over time?
The influence of MC- 391
LR on Dunaliella salina led to an increase in NPQ at the 15 h
Chlamydomonas
an increase in NPQ at the 15 h of exposure, after which its 392
value significantly decreased
394 NPQ remained lower
The high values 395 which high values?
this was not a general effect, according to fig 7, only 15 h for Chlamydomonas and final time for Dunaliella
395-398
unclear, please rephrase
1) photosynthetic activity was 401
significantly affected by MC-LR, with significant changes in all three parameters
not apparent from fig 5b 6B and 7C
404 was less affected by CYL
compared to MC?
statistic analysis in results presented in the figures compared each toxin with control, not CYL x MC, thus, results did not describe this
CYL. This cyanotoxin changes the parameter 404 non-photochemical quenching of chlorophyll fluorescence (NPQ), meaning that it acts by 405 inducing stress in the treated algae
but in fig 6 MC and CYL had similar effects
Similar effects of 410 toxin exposure
Which toxin? cyanotoxin?
This was 419 confirmed by the reduced production of ATP
This was indicated
422-425
please rephrase, unclear
methods
cyanotoxin concentrations
this study used a concentration of 1 µg/mL = 1000 ug / L
this is 1000 x over the reference values for drinking water
WHO guidelines values for MC-LR and CYL in drinking 451 water are 1 µg/L and 0.7 µg/L, respectively, but in aquatic environments, these toxins are 452 naturally occurring at concentrations 1-100 mg/L
It is rare that these toxins reach such high concentrations
please see CYL
Sci Total Environ. 2020 Oct 10;738:139807. doi: 10.1016/j.scitotenv.2020.139807.
MC also up to 25 ug / L
https://doi.org/10.1139/f2012-088
https://cdnsciencepub.com/doi/full/10.1139/f2012-088
Thus, the authors may be testing an extremely high concentration of toxins
496 4.5. Statistical Analysis
the authors should clarify if tests compared control vs MC and control vs MC
in many parts of the manuscript the authors make statements about MC x CYL, is there a statistical basis for this comparison?
conclusions
identify these species as bioindicators for presence of 509 cyanotoxins in water bodies
this is far from the results of this study and how to apply the findings in the environment
Reviewer 3 Report
The authors appropriately responded to my concerns. However, in adding table 1, which was a welcome addition, they left out any statistics designating if the inhibitory rates were statistically significant.
Round 3
Reviewer 3 Report
Thank you for your revisions. They appropriately addressed my cocncerns.